# Microtubule Dynamics Plays a Vital Role in Plant Adaptation and Tolerance to Salt Stress

**DOI:** 10.3390/ijms22115957

**Published:** 2021-05-31

**Authors:** Hyun Jin Chun, Dongwon Baek, Byung Jun Jin, Hyun Min Cho, Mi Suk Park, Su Hyeon Lee, Lack Hyeon Lim, Ye Jin Cha, Dong-Won Bae, Sun Tae Kim, Dae-Jin Yun, Min Chul Kim

**Affiliations:** 1Institute of Agriculture & Life Science, Gyeongsang National University, Jinju 52828, Korea; hj_chun@hanmail.net; 2Plant Molecular Biology and Biotechnology Research Center, Gyeongsang National University, Jinju 52828, Korea; dw100@hanmail.net (D.B.); misugip@hanmail.net (M.S.P.); 3Division of Applied Life Science (BK21 Four), Gyeongsang National University, Jinju 52828, Korea; scv5789@naver.com (B.J.J.); hmcho86@gnu.ac.kr (H.M.C.); leesuhyeon86@gmail.com (S.H.L.); dlafkrgus@gnu.ac.kr (L.H.L.); cdw3280@naver.com (Y.J.C.); 4Central Instrument Facility, Gyeongsang National University, Jinju 52828, Korea; bdwon@gnu.ac.kr; 5Department of Plant Bioscience, Life and Industry Convergence Research Institute, Pusan National University, Miryang 50463, Korea; stkim71@pusan.ac.kr; 6Department of Biomedical Science & Engineering, Konkuk University, Seoul 05029, Korea; djyun@konkuk.ac.kr

**Keywords:** salt stress, salt adaptation, proteomics, microtubules, tubulin

## Abstract

Although recent studies suggest that the plant cytoskeleton is associated with plant stress responses, such as salt, cold, and drought, the molecular mechanism underlying microtubule function in plant salt stress response remains unclear. We performed a comparative proteomic analysis between control suspension-cultured cells (A0) and salt-adapted cells (A120) established from *Arabidopsis* root callus to investigate plant adaptation mechanisms to long-term salt stress. We identified 50 differentially expressed proteins (45 up- and 5 down-regulated proteins) in A120 cells compared with A0 cells. Gene ontology enrichment and protein network analyses indicated that differentially expressed proteins in A120 cells were strongly associated with cell structure-associated clusters, including cytoskeleton and cell wall biogenesis. Gene expression analysis revealed that expressions of cytoskeleton-related genes, such as *FBA8*, *TUB3*, *TUB4*, *TUB7*, *TUB9*, and *ACT7*, and a cell wall biogenesis-related gene, *CCoAOMT1*, were induced in salt-adapted A120 cells. Moreover, the loss-of-function mutant of *Arabidopsis* *TUB9* gene, *tub9*, showed a hypersensitive phenotype to salt stress. Consistent overexpression of *Arabidopsis* *TUB9* gene in rice transgenic plants enhanced tolerance to salt stress. Our results suggest that microtubules play crucial roles in plant adaptation and tolerance to salt stress. The modulation of microtubule-related gene expression can be an effective strategy for developing salt-tolerant crops.

## 1. Introduction

Plant adaptation to environmental stress is regulated by cascades of molecular networks, including stress perception, signal transduction, metabolic adjustment, and the regulation of stress-responsive gene expressions, to reestablish cellular homeostasis, such as osmotic and ionic homeostasis, and protect proteins and cell membranes by using heat shock proteins (Hsps), chaperones, late embryogenesis abundant (LEA) proteins, osmoprotectants, and free-radical scavengers [1]. Plant cells have adapted to salt stress by changing cell wall composition [2,3]. Extensin, a significant cell wall glycoprotein, is cross-linked with phenolics by reactive oxygen species (ROS) accumulation to stiffen the cell wall when plant cells are exposed to salt stress [2]. The RhEXP4, expansin A4 of rose, overexpressing *Arabidopsis* plants show increased seed germination, root growth, and several lateral roots under salt stress conditions [4]. A higher pectin content in root tips enhances plant tolerance to salt stress by increasing root growth compared with the cell wall composition in two soybean cultivars [5]. The dysfunction of *Arabidopsis* AtCSLD5, a pectin biosynthesis enzyme in *sos6* (salt overly sensitive 6) mutant, enhances plant sensitivity to salt stress [6]. The *Arabidopsis* CC1 and CC2 proteins, along with cellulose synthases, interact with microtubules (MTs) and are essential for seedling growth under salt stress conditions [7]. However, the molecular mechanisms of changing cell wall dynamics by salt stress, including signal transduction and cell wall integrity pathways, remain unclear.

The plant cytoskeleton comprises the systemic polymers between actin filaments and MTs [8,9]. MTs form heterodimers by polymerization between α-tubulin and β-tubulin [10]; also, actin filaments polymerize to form filamentous structures by G-actin [11]. The adaptive mechanisms of the plant cytoskeleton to salt stress are varied by organization, dynamics, and cellular processes [12,13]. MTs play essential roles in the cell cycle, cell growth, and stress response by forming highly dynamic polymers [14,15]. Additionally, MT depolymerization and reorganization are essential for enhancing plant tolerance to salt stress [16]. In plants, response to salt stress, calcium ions, abscisic acid (ABA), and ROS as signaling molecules are associated with cortical MT array organization [17,18,19]. Cytosolic-increased calcium induces MT depolymerization by regulating calcium channels in the salt stress response [18]. The plant hormone, ABA, influences the organization and stability of cortical MTs [18]. ABA promotes the ectopic derivative of root cells by depolymerizing and reorganizing cortical MTs and activating MT depolymerization in guard cells during stomatal closure [20,21]. ROS induces the MTs’ reorganization through MT disassembly and the formation of irregular MT polymers [19]. When ROS homeostasis is collapsed by salt stress, tubulin forms a modified structural state by assembling non-typical tubulin structures [22]. Propyzamide-hypersensitive 1 (PHS1), a mitogen-activated protein kinase phosphatase, phosphorylates α-tubulin and elevates MT depolymerization to salt stress [23]. When plants are exposed for a long period to salt stress, cortical MT reorganization is induced by the depolymerization and reassembly of MT networks [17]. The MT-associated proteins, 65-1 and MAP65-1, facilitate MT polymerization and bundling, enhance MT-stabilizing activity, and expedite cortical MT recovery by binding phosphatidic acid under salt stress conditions [24]. Although the role of MTs during plants’ responses to salt stress has been much studied, the mechanism of the actin cytoskeleton is less understood. The actin cytoskeleton leads to assembly and bundle formation in response to a short period of salt stress; however, long-term exposure to salt stress or exposure to high salt stress induces the disassembly of the actin cytoskeleton [25]. Salt stress regulates the cellular process of actin dynamics via the salt overly sensitive (SOS) pathway and calcium signaling [26]. The *arp2* (actin-related protein 2) mutant showed a hypersensitive phenotype to salt stress by increasing mitochondria-dependent [Ca^2+^]_cyt_ levels [27].

Salt stress affects the expression levels of MT-associated genes and proteins. The loss-of-function mutants of prefoldin subunits 3 (PFD3) and PFD5 showed a hypersensitive phenotype to salt stress by decreasing expression levels of α-tubulin and β-tubulin [28]. The 26S proteasome degrades MT-associated protein SPIRAL1 (SPR1)-stabilizing MT in response to salt stress [16]. Proteomics analyses reveal that the plant adaptation to salt stress is associated with complex networks of protein expression and post-translational modifications [29,30,31]. Functional profiling of various proteins by a comparative proteomic approach has made it possible to characterize essential proteins involved in salt tolerance in various plant species, including *Thellungiella halophila* [32], *Halogetong lomeratus* [33], *Tangut Nitraria* [34], canola [35], sesame [36], and rice [37]. Adaptation to salt stress is a congested process in the whole plant and cellular levels and needs to adjust the transcription of various genes that trigger protein profile change [38,39]. Thus, quantitative analysis of expressed proteins by proteomics is valuable for understanding the molecular mechanisms underlying plant adaptation and tolerance to salt stress.

Our previous metabolite profiling study using salt-adapted *Arabidopsis* callus suspension-cultured cells reveals that various cellular processes, including cell wall thickening, play essential roles in plant salt adaptation [40]. In this proteomics study, we revealed that major differentially expressed proteins (DEPs) identified from salt-adapted cells were functionally associated with cytoskeleton and cell wall biogenesis. Structural and morphological changes of plant cells mediated by cytoskeleton and cell wall biogenesis functions are vital for adaptation and tolerance to salt stress.

## 2. Results

### 2.1. Morphological Features of Salt-Adapted Callus Suspension-Cultured Cells

Plants exhibit growth inhibition and impediment of tissue development in response to salt stress because of a deficit of cell wall extensibility [41]. When we compared morphologies between control cells (A0) and salt-adapted cells (A120; adapted to 120 mM NaCl), we observed that the A120 cells showed distinct morphological changes compared with A0 cells, including spherical or ellipsoidal and isodiametric shapes (Figure 1a). Additionally, newly divided A120 cells stuck together in small clumps. Vacuole size and the cytoplasmic volume in A120 cells were significantly reduced compared with those in A0 cells (Figure 1a). These data suggested that plant suspension cells have changed their morphology to adapt to long periods of salt stress. To understand the molecular mechanism underlying cell morphology changes during salt adaptation, we identified DEPs in salt-adapted A120 cells by proteomics analysis. Additionally, we characterized their biological functions by molecular genetic analysis using *Arabidopsis* mutants and transgenic rice plants (Figure 1b).

### 2.2. Overview of Proteomic Profiles in Salt-Adapted Cells

Crude proteins were extracted from A0 and A120 cells grown in normal media (for A0 cells) and saline media with 120 mM NaCl (for A120 cells) for 8 days after subculture using the trichloroacetic acid/acetone/phenol extraction protocol [42] and quantified using a 2D-Quant Kit (GE Healthcare, Waukesha, WI, USA). Representative two-dimensional gel electrophoresis (2-DE) images from three biological replicates of A0 and A120 cells are displayed in Figure 2. With a cut-off point as a *p*-value of <0.05 for the differential expression between A0 and A120 cells, 50 DEP spots were identified by matrix-assisted laser desorption/ionization time-of-flight mass spectrometry (MALDI-TOF/TOF MS) (Table 1). When comparing expression levels in A120 cells with those in A0 cells, we identified 45 induced spots and 5 reduced spots in A120 cells (Table 1). Fifty DEPs identified in A120 cells were classified into functional categories based on gene ontology (GO) analysis using the PANTHER program (http://pantherdb.org) (Figure 3). The DEPs were included in “binding” (42.6%), “catalytic activity” (42.6%), “structural molecule activity” (8.5%), and “translation regulator activity” (6.4%) categories in the molecular function (Figure 3a, red color). In the biological process, DEPs were included in five categories, which are “biological regulation” (1.9%), “cellular process” (55.6%), “localization” (1.9%), “metabolic process” (33.3%), and “response to stimulus” (7.4%) (Figure 3a, green color). The DEPs were included in “cellular, anatomical entity” (46.6%), “intracellular” (48.3%), and “protein-containing complex” (5.2%) categories in the cellular component (Figure 3a, black bar). In the analysis of protein class, the two largest proportions of DEPs belonged to the “metabolite interconversion enzyme” (41.5%) and “cytoskeletal protein” (14.6%) classes (Figure 3b). Above these, DEPs were included in “calcium binding protein” (4.9%), “chaperone” (12.2%), “gene-specific transcriptional regulator” (2.4%), “nucleic acid metabolism protein” (2.4%), “protein modifying enzyme” (7.3%), “protein-binding activity modulator” (2.4%), and “translational protein” (9.8%) classes (Figure 3b). In the analysis of pathway class, the largest proportion of DEPs belonged to the “cytoskeletal regulation by Rho GTPase” (30.8%) class (Figure 3c). Above these, DEPs were associated with “apoptosis signaling pathway” (7.7%), “cell cycle” (7.7%), “de novo purine biosynthesis” (7.7%), “fructose galactose metabolism” (7.7%), “glycolysis” (7.7%), “S-adenosylmethionine biosynthesis” (15.4%), “TCA cycle” (7.7%), and “ubiquitin-proteasome pathway” (7.7%) (Figure 3c). Thus, our results suggested that critical cellular changes during plant adaptation to salt stress were related to cytoskeletal regulation and metabolite processes.

### 2.3. Functional Network Analysis of Differentially Expressed Proteins

To understand the biological functions and modes of action of 50 DEPs in plant salt adaptation, we analyzed putative physical interactions of DEPs using the Cytoscape software platform (https://cytoscape.org/) (accessed on 1 April 2021) and the IntAct database (https://www.ebi.ac.uk/intact/) (accessed on 1 April 2021) (Figure 4). The Cytoscape with large databases of protein–protein, protein–DNA, and genetic interactions is a powerful software for studying the prediction of a physical interaction network in model organisms [43]. Out of 50 DEPs, the physical interactions of 34 DEPs were identified from this analysis. The largest cluster was the “cell structure-associated cluster,” including 12 DEPs (red ellipse) in the functional network. The proteins in this cluster were mainly involved in the regulation of cell structures, including both cytoskeleton functions, such as actin filaments (ACT7, ADF3, and FBA8) and MTs (TUB3, TUB4, and TUB9), and secondary cell wall biogenesis (CCoAOMT1) (Figure 4). Even though TUB7 and PCAP1 proteins were highly induced in A120 cells (Table 1), they were not identified in functional network analysis. This is probably due to the lack of physical interaction information identified so far. PCAP1, also known as MT-destabilizing protein 25 (MDP25), functions as a negative regulator in hypocotyl cell elongation [44]. Additionally, other DEPs physically interacted with various functional proteins clustered in the ROS-associated cluster (CTIMC and ANNAT1; green ellipse), drought- and ABA-associated cluster (GRF3 and RBG2; purple ellipse), temperature-associated cluster (HSP70-1, HSP70-9, HOP2, and CSP2; orange ellipse), and transcriptional/translational system-associated cluster (PBD1, RPN11, PAP1, and CPN20; gray box) (Figure 4). The connectivity of protein interaction networks suggested that significant cellular and molecular changes in plant adaptation to salt stress might be associated with the plant cytoskeleton and cell wall biogenesis, affecting cell structure changes.

### 2.4. Expression Patterns of Cytoskeleton-Related Genes in Salt-Adapted Cells

To confirm the results of proteomics and bioinformatics analyses suggesting the crucial roles of cell structure-related proteins in plant salt adaptation (Table 1 and Figure 4), we tested the expressions of 12 genes encoding DEPs. This belonged to the cytoskeleton and cell wall biogenesis functions between A0 and A120 cells using quantitative real-time PCR (qRT-PCR). We also included two cytoskeleton-related genes, TUB7 and PCAP1, in the gene expression analysis.

The expression of cytoskeleton-related genes, including AT3G52930 (FBA8), AT5G62770 (TUB3), AT5G44340 (TUB4), AT2G29550 (TUB7), AT4G20890 (TUB9), and AT5G09810 (ACT7), and cell wall-related gene, AT4G34050 (CCoAOMT1), was significantly induced in A120 cells (Figure 5). However, the expression of other genes, such as AT4G13940 (SAHH1), AT4G20260 (PCAP1), and AT4G10480 (NACα4), in this cluster was decreased in A120 cells. Those of AT2G47470 (PDIL2-1), AT1G21750 (PDIL1-1), AT3G62030 (CYP20-3), and AT5G59880 (ADF3) were similar in A0 and A120 cells (Figure 5). Among 14 genes tested, FBA8, TUB3, TUB4, TUB7, TUB9, ACT7, and CCoAOMT1 were induced in mRNA (Figure 5) and protein levels (Table 1) in salt-adapted A120 cells. Although TUB7 was not identified in functional network analysis (Figure 4), its mRNA was more abundant in A120 cells (Figure 5). Since four TUB genes, TUB3, TUB4, TUB7, and TUB9, were induced in the protein and mRNA levels in A120 cells, MT-related proteins play essential roles in plant adaptation to salt stress.

### 2.5. The Effect of the Loss-of-Function β-Tubulin Genes in Salt Stress Response

Recent evidence indicates that the regulation of MTs’ destabilization and reorganization is essential for plant adaptation to salt stress [17,45,46]. Furthermore, the largest protein family among the cell structure-associated cluster is related to the β-tubulin family proteins, including TUB3, TUB4, TUB7, and TUB9 (Table 1 and Figure 4). To characterize the physiological functions of β-tubulin in salt stress responses, we isolated T-DNA insertion mutants of *Arabidopsis* β-tubulin genes (*tub3*, SALK_073132; *tub4*, SALK_204506; *tub7*, SALK_026797; *tub9*, SALK_015876). Wild-type (WT, ecotype Col-0) and four *tub* mutant plants were grown on MS medium for 5 days and then transferred to the soil to test mutant phenotypes under salt stress conditions. After 9 days, we supplied water containing 130 mM NaCl to the soil once a week for 4 weeks. The *tub4* mutants displayed strongly tolerant phenotypes, such as enhanced plant height and late wilting of leaves, to salt stress compared to WT plants (Figure 6b). In contrast, *tub9* mutants were hypersensitive to salt stress with a quickly wilting phenotype compared with WT plants (Figure 6d). Furthermore, *tub3* and *tub7* mutants showed similar phenotypes, including plant height and wilting of leaves, to WT plants under salt stress conditions (Figure 6a,c). These results suggested that both TUB4 and TUB9 play significant roles in plant adaptation and tolerance to salt stress, but their mode of function differs.

### 2.6. The Effect of TUB9 Overexpression in Rice during Salt Stress

The hypersensitive phenotype of *Arabidopsis tub9* mutants to salt stress suggests that the overexpression of *Arabidopsis TUB9* gene can enhance crop tolerance to salt stress. To confirm this, we generated transgenic rice plants overexpressing *Arabidopsis TUB9* gene under the control of the *CaMV 35S* promoter (*TUB9*-OX). The *Arabidopsis TUB9*-OX construct was transformed into rice (“Ilmi” cultivar) embryogenic callus, and three independent *TUB9*-OX T_1_ lines were selected by hygromycin B resistance and RT-PCR analysis. Under normal conditions, *TUB9*-OX transgenic plants were shorter than WT plants (Figure 7a). Besides plant height, other morphological phenotypes of TUB9-OX transgenic plants were comparable with WT plants. Ten-day-old WT and *TUB9*-OX transgenic plant seedlings were transferred into MS liquid media containing 120 mM NaCl. After 7 days of salt treatment, salt-treated WT and *TUB9*-OX transgenic plants were recovered in liquid MS medium without NaCl for 10 days. The *TUB9*-OX transgenic plants had greener leaves and higher heights than WT plants (Figure 7b). The number of rice transgenic plants with green leaves in WT and *TUB9*-OX transgenic lines was counted in the recovery stage after salt treatment to calculate the survival rate. The survival rate of *TUB9*-OX transgenic plants was approximately 40%; however, most WT plants had no green leaves (Figure 7b,c). These results suggested that *Arabidopsis TUB9* gene functions as a positive regulator in plant adaptation to salt stress and can enhance plant tolerance to salt stress.

## 3. Discussion

Salt stress disrupts cell division in leaves and roots through various cellular mechanisms, such as calcium ion, ROS, and ABA-dependent responses [47]. The changes in cellular morphology, such as cell proliferation and cell expansion, are essential for plant adaptation and tolerance to salt stress [47,48]. However, cellular and molecular mechanisms of morphological changes during salt adaptation have not been well elucidated. This study demonstrated that salt-adapted A120 cells showed morphological changes, such as spherical or ellipsoidal and isodiametric shapes, compared with control A0 cells (Figure 1a). Results of GO and network analysis using proteomics data showed that many DEPs identified from salt-adapted cells were associated with regulating cell structures, including cytoskeleton and cell wall biogenesis (Figure 3 and Figure 4). Moreover, our gene expression and molecular genetic analyses revealed that β-tubulin family proteins play positive and negative roles in plant adaptation and tolerance to salt stress (Figure 6 and Figure 7). Our results suggest that β-tubulin MTs are vital components in modulating plant adaptation and tolerance to salt stress.

### 3.1. Molecular Functions of Differentially Expressed Proteins in Salt-Adapted Cells

This study elucidated the molecular mechanisms underlying plant adaptation to prolonged salt stress by comparative proteomics between control and salt-adapted cells. The previous proteomics studies conducted using suspension cells demonstrated that molecular mechanisms of suspension cells in salt stress response are complicated but similar to those studied at the whole plant level [49,50]. Using proteomics, we identified 50 DEPs, including 45 up-regulated and 5 down-regulated proteins, in salt-adapted cells compared with control cells (Table 1). Functional network analysis revealed that the identified DEPs were included in various functional clusters, but many of them in cell structure-associated clusters, including cytoskeleton and cell wall biogenesis functions (Figure 4).

#### 3.1.1. Cell Structure-Associated Cluster

The plant cell surface comprises the cell wall, plasma membrane, and cytoskeleton [41]. Plant cytoskeletons play essential functions in plant tolerance and survival to salt stress [17,18]. Many up-regulated proteins in A120 cells were MTs and actin filament-related proteins (Table 1 and Figure 4). The ACT7 (AT5G09810), ADF3 (AT5G59880), and FBA8 (AT3G52930) proteins were involved in the actin cytoskeleton. Actin cytoskeletons are composed of two classes, which are vegetative (ACT2, ACT7, and ACT8) and reproductive (ACT1, ACT3, ACT4, ACT11, and ACT12). ACT7 transcription is high in vegetative organs and induced by auxin [51]. The act11 mutant decreases pollen germination and increases pollen tube growth by increasing the actin turnover rate [52]. However, the loss-of-function ACT2 mutant vegetative class affects root hair growth but is not complemented by overexpressing ACT7, even if they are of the same classes [53]. Additionally, ACT7 physically interacts with ACT1, ACT11, ACT12, actin-depolymerizing factor 6 (ADF6), and actin-interacting protein 1-2 (AIP1-2) (Figure 4). ADF3 (actin-depolymerizing factor 3) depolymerizes F-actin and acts as a crucial regulator in plant defense response to biotic stress [54]. Abiotic stresses also regulate the protein and gene expression of ADFs. OsADF proteins in rice leaves are highly accumulated because of drought stress [55]. OsADF3 protein is induced by salt stress in two rice cultivars (Oryza sativa L. cv. Nipponbare and Oryza sativa L. cv. Tainung 67) [56,57]. FBA8, which encodes fructose-bisphosphate aldolase 8, is involved in actin polymerization and various abiotic stress responses, such as salt, drought, ABA, and temperature stresses [50,58]. MT dynamics, polymerization and depolymerization, are necessary for cellular processes of plant tolerance and adaptation to salt stress [17]. Our results revealed that TUB3, TUB4, TUB7, and TUB9 proteins, involved in MT depolymerization and reorganization, play vital roles in plant adaptation and tolerance to salt stress (Table 1 and Figure 4). It was also reported that TUA6 (α-chain tubulin 6) and TUB2 (β-chain tubulin 2) proteins are highly expressed in Arabidopsis roots in response to salt stress [59].

CCoAOMT1 (AT4G34050), encoding caffeoyl-coA o-methyltransferase 1, plays an essential role in lignin biosynthesis and salt stress response [60]. *ccoaomt1* mutants showed a hypersensitive phenotype to salt and drought stresses [60,61]. SAHH1 (AT4G13940), encoding S-adenosyl-L-homocysteine hydrolase 1, is involved in the interaction between cytokinin and DNA methylation. Protein disulfide isomerase-like (PDIL) protein 1-1 (PDIL1-1, AT1G21750), which plays essential roles in ER trafficking, is associated with the response to salt stress. The loss-of-function mutant of PDIL2-1 (AT2G47470) disrupts pollen tube growth by delaying embryo development [62]. In addition, protein disulfide isomerase genes in maize are induced by abiotic stress, such as salt, drought, ABA, and H_2_O_2_ [63]. Cyclophilin CYP20-3 (ROC4, AT3G62030), which plays an important role in redox regulation, is involved in salt stress response [64]. CYP20-3 protein composes the sophisticated and reticular connective networks with FBA8, SAHH1, CCoAOMT1, PDIL1-1, and PDIL2-1 (Figure 4). However, nascent polypeptide-associated complex (NAC) alpha subunit family protein (NACα4 and AT4G10480) is involved in the cell structure-associated cluster (Figure 4). NACα4 function is not well known; however, the NAC complex plays a vital role in abiotic stress responses, such as drought and salt in barley [65].

#### 3.1.2. ROS-Associated Cluster

Plants depend on cellular signaling and pathways via the reestablishment of ROS homeostasis in salt stress adaptation [66]. CTIMC (AT3G55440), which encodes triosephosphate isomerase (TPI), plays an essential role in redox regulation and is induced in response to salt stress by reactive carbonyl species (RCS). CTIMC protein forms a complex network in abiotic stress signaling through redox regulation by ROS-related proteins, including protein detoxification (DTX) proteins, L-type lectin receptor kinases 32 (LECRK32), plasma membrane intrinsic protein 1-5 (PIP1-5), nitrate transporter 1.1 (NPF1.1), cysteine-rich RLK 2 (CRK2), and acyl-lipid desaturase 2 (ADS2) (Figure 4). ANNAT1 (AT1G35720), annexin protein, has peroxidase activity and is involved in various abiotic stresses, such as salt, drought, and ABA. The *annat1* mutant showed tolerance to salt and drought stress by regulating ABA and proline biosynthesis [67]. ANNAT1 protein forms a complex network in various signaling pathways by controlling ROS-related proteins, including NPF6.1, 7.2, 8.2, 8.3, ADS2, respiratory burst oxidase homolog D (RBOHD), LysM Receptor-Like Kinase1 (CERK1), PIN5, cyclic nucleotide-gated channels 17 (CNGC17), proline transporter 1 (PROT1), and sugar transporter 7 (SWEET7) (Figure 4).

#### 3.1.3. Drought- and ABA-Associated Cluster

Growth regulating factor 3 (GRF3) influences plant tolerance to drought stress and organ growth by increasing leaf size. Glycine-rich RNA-binding protein 2 (RGB2/GRP2) affects seed germination in an ABA-independent manner under salt stress. GRF2 and RGB2 proteins are related to various plant stress-associated proteins, such as CBL-interacting protein kinases (CIPK12), cyclin-B 2-2 (CYCB2-2), ABA-responsive element-binding protein 3 (AREB3/DPBF3), cytosolic invertase 1 (CINV1), and cyclin-H 1-1 (CYCH1-1) (Figure 4).

#### 3.1.4. Temperature-Associated Cluster

Heat shock 70 kDa protein 1 (HSP70-1), a key component in protein folding, plays a vital role in stomatal closure and seed germination and response to ABA stress. Mitochondrial HSP70-9 protein is involved in iron–sulfur protein biogenesis. Cold shock protein 2 (CRP2) protein plays different roles as a negative regulator in response to cold stress and as a positive regulator in salt stress response. Hsp70-Hsp90 organizing protein 2 (HOP2) influences plant adaptation to prolonged heat stress. These four proteins compose functional networks via physical interaction with chaperon regulators, including heat stress transcription factor A-1 (HSF1A), suppressor of G2 allele of skp1 (SGT1) homolog B (SGT1B), Bcl-2-associated athanogene 3 (BAG3) and 5 (BAG5), and cytokinin response factor 3 (CRF3) (Figure 4).

#### 3.1.5. Transcriptional/Translational System-Associated Cluster

Proteasome subunit beta type 2-A (PBD1) and regulatory particle non-ATPase 11 (RPN11) play an important role in the plant ubiquitin–proteasome system via protein degradation and stabilization. Plastid-lipid-associated protein 1 (PAP1) contributes to the protection of photosystem II (PSII) and in response to ABA stress. Chloroplast co-chaperonin 20 (CPN20) acts as a negative regulator in ABA signaling. These four proteins compose functional networks via physical interaction with protein stability- or gene transcription-related proteins, including 26S proteasome regulatory subunit 4 homolog A (RPT2A), RPN1A, trithorax-related protein 5 (ATXR5), importin alpha isoform 6 (IMPA6), calmodulin-like protein 18 (CML18), sensitive to proton rhizotoxicity 2 (STOP2), authentic response regulator 14 (ARR14), budding uninhibited by benzymidazol 3.3 (BUB3.3), and WRKY17 (Figure 4).

### 3.2. The Role of Microtubules in Plant Adaptation and Tolerance to Salt Stress

MTs are fixed in the plasma membrane and composed of a greater part of plant interphase arrays [9,15]. The cortical MT arrays are involved in plant response to various abiotic stresses, especially salt stress [9,18]. Plants increase salt tolerance by regulating depolymerization and reorganization of the cortical MTs [48]. MAP65-1 acts as a positive regulator in plant salt tolerance by promoting cortical MT reorganization [68]. Calcium ions reorganize the damage of MT arrays in the salt stress response of plant cells [17]. The loss-of-function sos3, a calcium sensor in the salt stress response, mutant shows hypersensitivity to salt stress due to the irregular organization of MTs [26]. Plants with salt-susceptible phenotypes have a lower concentration of calcium ions than that of salt-tolerant plants [69]. Our proteomic analysis showed that the four β-tubulin family proteins, including TUB3, TUB4, TUB7, and TUB9, were induced in salt-adapted A120 cells compared with control A0 cells (Table 1 and Figure 4). Additionally, the mRNA levels of TUB3, TUB4, TUB7, and TUB9 genes were higher in A120 cells than in A0 cells (Figure 5). Our results suggest that the elevation of β-tubulin mRNAs and protein levels can affect MT functions and enhance plant adaptation to salt stress. In our molecular genetic analysis, the loss-of-function tub4 mutant showed enhanced tolerance to salt stress. In contrast, the tub9 mutant was more hypersensitive than WT plants (Figure 6). The overexpression of TUB9 in rice plants enhanced the plant’s tolerance to salt stress (Figure 7). Interestingly, tub4 and tub9 mutant plants showed opposite phenotypes in response to transiently applied salt stress, even though TUB4 and TUB9 protein levels were higher in cells that have adapted to salt stress for a long time. These results suggest that TUB4 and TUB9 proteins play different roles in plant responses to short-term and long-term salt stresses. It was also reported that short-term and long-term salt stress have different effects on the actin filament assembly and disassembly [25]. It would be worthwhile to dissect the biological functions of TUB4 and TUB9 in plant adaptation and tolerance to salt stress in further studies.

Altogether, our results suggest that β-tubulin proteins play different roles in plant adaptation and tolerance to salt stress by regulating MT depolymerization and reorganization. Therefore, changes in MT dynamics in plant cells would be essential for cellular processes to enhance the adaptation and tolerance to salt stress. Furthermore, morphological changes in salt-adapted suspension cells are at least partly due to the changes in MT dynamics.

## 4. Materials and Methods

### 4.1. Growth Conditions of Callus Suspension Cells

Salt-adapted callus suspension cells were generated from *Arabidopsis thaliana* (Col-0 ecotype) roots as described in detail in a previous study [40]. Callus suspension cells were maintained at 23 °C in the dark with gentle shaking (140 rpm).

### 4.2. Proteomic Profiling Using Two-Dimensional Gel Electrophoresis

Total protein was isolated from 5 g of A0 and salt-adapted cells (A120) using trichloroacetic acid/acetone/phenol extraction protocol described in detail in a previous study [42]. Total soluble proteins were quantified using the 2D-Quant Kit (Amersham Biosciences Europe GmbH, Freiburg, Germany). Two-dimensional gel electrophoresis was performed with Protean IEF cell (Bio-Rad, Hercules, CA, USA) for the first-dimensional isoelectric focusing using immobilized pH gradient strips (24 cm, pH 4–7; Bio-Rad Laboratories, Hercules, CA, USA), and with the Protean Xi-II Cell system (Bio-Rad Laboratories, Hercules, CA, USA) for the second-dimensional sodium dodecyl sulfate–polyacrylamide gel electrophoresis. After Coomassie brilliant blue staining, gel images were taken using a GS-800 Imaging Densitometer Scanner (Bio-Rad Laboratories, Hercules, CA, USA) and analyzed using PDQuest v.7.2.0 (Bio-Rad Laboratories, Hercules, CA, USA). All experiments were performed in three independent biological replicates, and the volume of each spot was detected and normalized to a relative density. Proteins showing a statistically significant difference (*p* < 0.05) between A0 and A120 cells were identified. For protein identification, differential protein spots visualized in the gel were excised and subjected to in-gel digestion as described previously [42]. Protein identification was performed by MALDI-TOF/TOF MS using the ABI 4800 Plus TOF-TOF Mass Spectrometer (Applied Biosystems, Framingham, MA, USA). Fifty proteins were identified, of which peptide and fragment mass tolerance was fixed at 100 ppm. The high confidence interval displayed statistically reliable search scores (more than 95% confidence) corresponding to protein’s experimental isoelectric point (pI) and molecular weight.

### 4.3. Bioinformatics Analysis

The functional classification of DEPs identified in proteomics was performed using the PANTHER classification system (http://www.pantherdb.org/) (accessed on 3 April 2021). We used network-based enrichment by Cytoscape software platform to forecast physical interactions of DEPs (https://cytoscape.org/) (accessed on 1 April 2021) using the IntAct database (https://www.ebi.ac.uk/intact/) (accessed on 1 April 2021).

### 4.4. Analysis of Quantitative Real Time PCR (qRT-PCR)

Total RNA was extracted from A0 and A120 cells using the RNeasy Plant Kit (Qiagen, Valencia, CA, USA) following the manufacturer’s protocol. To remove genomic DNA contaminants, extracted RNA was treated with DNaseI (Thermo Fisher Scientific, Waltham, MA, USA). One µg of total RNA was used the first strand of cDNA synthesis using a cDNA synthesis kit (Invitrogen, Carlsbad, CA, USA), according to the manufacturer’s protocol.

The qRT-PCR analysis was performed using the QuantiMix SYBR (PhileKorea, Seoul, Korea), and the relative values of indicated gene expression were automatically calculated using the CFX96 real-time PCR detection system (Bio-Rad Laboratories, Hercules, CA, USA) by applying normalization of the expression of *UBQ10*. The qRT-PCR was performed using the following conditions: 50 °C for 10 min, 95 °C for 10 min; followed by 50 cycles at 95 °C for 15 s, 60 °C for 15 s, and 72 °C for 15 s. The gene specific primers in qRT-PCR analysis are listed in Appendix A.

### 4.5. Plant Materials and Growth Conditions

*Oryza sativa* L. (“Ilmi” cultivar) and *Arabidopsis thaliana* (Col-0 ecotype) plants were used in all experiments. Rice plants were grown under natural light conditions in a greenhouse at 25–30 °C. The *tub3* (SALK_073132), *tub4* (SALK_204506), *tub7* (SALK_026797), and *tub9* (SALK_015876) mutants were obtained from Arabidopsis Biological Resource Center (https://www.arabidopsis.org/) (accessed on 24 February 2014). *Arabidopsis* plants were grown in a growth chamber in long-day conditions (16 h light/8 h dark) at 23 °C.

### 4.6. Generation of Transgenic Rice Plants

To generate the transgenic rice plants overexpressing the *Arabidopsis TUB9* gene, we cloned the full-length cDNA (1335 bp) of the *Arabidopsis*
*TUB9* gene into *pH2GW7* vector under the control of *CaMV 35S* promoter. The *TUB9-*OX construct was introduced into *Agrobacterium tumefaciens* (LBA4404) by electroporation. We used a modified version of the general rice-transformation protocol [70]. Transgenic *TUB9*-OX (T1) plants were selected on MS medium containing hygromycin B and then transferred to soil and allowed to self-pollinate.

### 4.7. Salt Stress Treatment

In *Arabidopsis*, 5-day-old WT seedlings, *tub3*, *tub4*, *tub7*, and *tub9* plants grown on MS media were transferred to soil. After 9 days, we supplied water containing 130 mM NaCl to the soil once a week for 4 weeks. Photographs of each representative of 12–16 individual plants were taken to analyze plant phenotypes. In rice, 10-day-old WT seedlings and *TUB9*-OX plants germinated in MS media containing hygromycin B were transferred into MS liquid medium with 120 mM NaCl. After 7 days, plants were recovered in MS solution without NaCl for 10 days. Photographs were taken to represent 8–10 individual plants to analyze plant phenotypes.

### 4.8. Statistical Analyses

Statistical analyses, including Student’s *t*-test, were performed using Excel 2010. qRT-PCR analysis was performed in three independent biological replicates, and the average values of 2^ΔΔ^CT were used to determine expression differences. Data were indicated as means ± standard deviation (SD). Error bars indicate SD.

## 5. Conclusions

This study suggests that the morphological changes of plant cells are an essential cellular process for adaptation to prolonged salt stress. We revealed that various protein families involved in various cellular processes play a role in salt adaptation response using proteomic analysis. Furthermore, gene expression and molecular genetic analyses demonstrated that β-tubulin proteins play an important role in plant adaptation and tolerance to salt stress. Altogether, our results suggest that the dynamics of depolymerization and reorganization of tubulin MTs play critical roles in plant adaptation to salt stress.

## Figures and Tables

**Figure 1 ijms-22-05957-f001:**
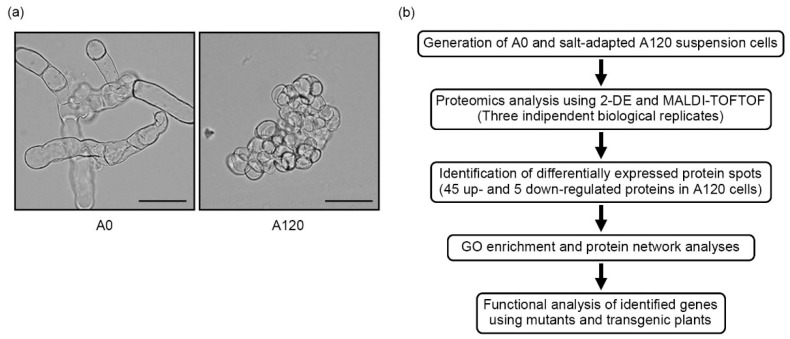
Characterization of *Arabidopsis* salt-adapted cells. (**a**) The morphological phenotype of control suspension cells (A0) grown in normal MS medium and salt-adapted cells (A120) grown in high salt MS medium with 120 mM NaCl. The photograph was taken using a microscope after 2 weeks of subculture. Scale bars indicate 50 μm. (**b**) Experimental scheme of proteomic and functional analyses.

**Figure 2 ijms-22-05957-f002:**
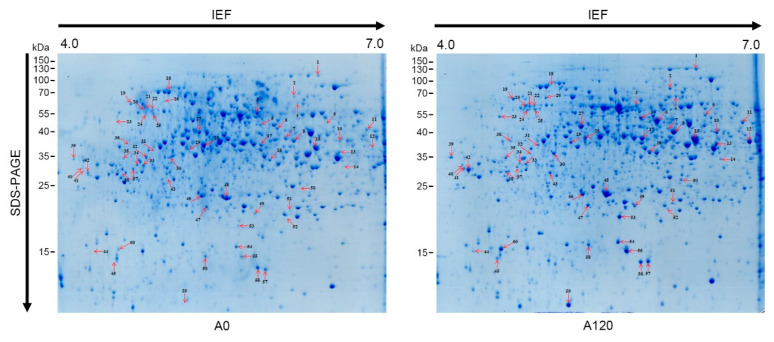
Proteome analysis of control A0 and salt-adapted A120 cell lines. The 50 differentially expressed protein (DEP) spots (45 up- and 5 down-regulated in A120 cells) were identified by 2-DE and MALDI-TOF/TOF analyses.

**Figure 3 ijms-22-05957-f003:**
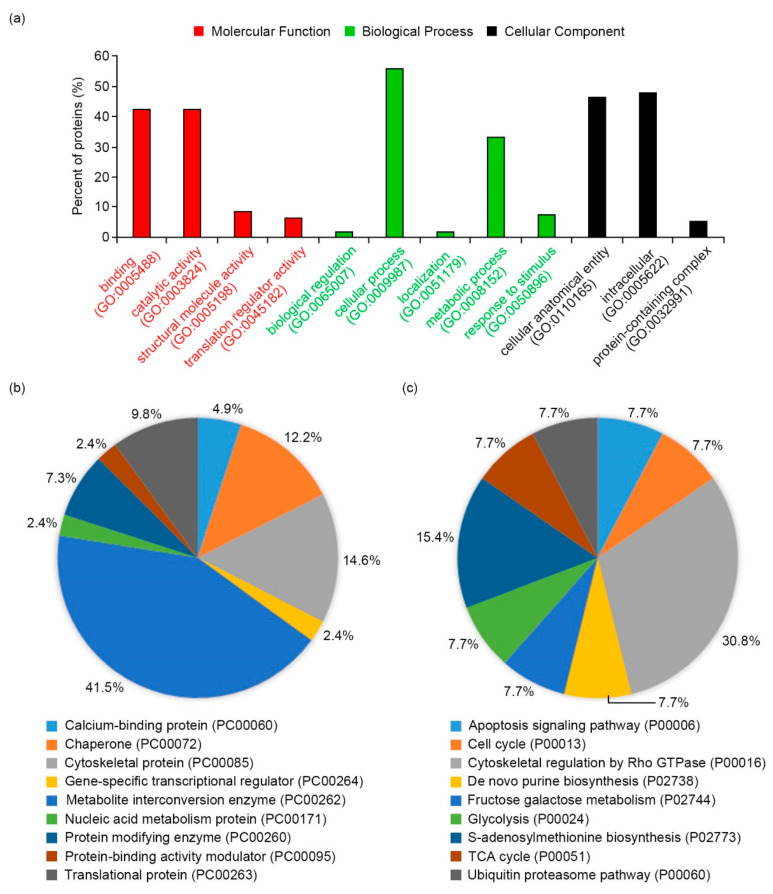
Functional classification of DEPs using the PANTHER database. (**a**) The bar chart of molecular function (red), biological process (green), and cellular components (black) represented is based on the PANTHER GO analysis. (**b**) Bar chart of the PANTHER protein classes in DEPs. (**c**) Bar chart of the PANTHER pathway classes in DEPs. The input percentage was calculated on the basis of the number of proteins mapped to the GO term divided by all protein numbers in the lists of DEPs (*Arabidopsis thaliana* IDs from the NCBI database).

**Figure 4 ijms-22-05957-f004:**
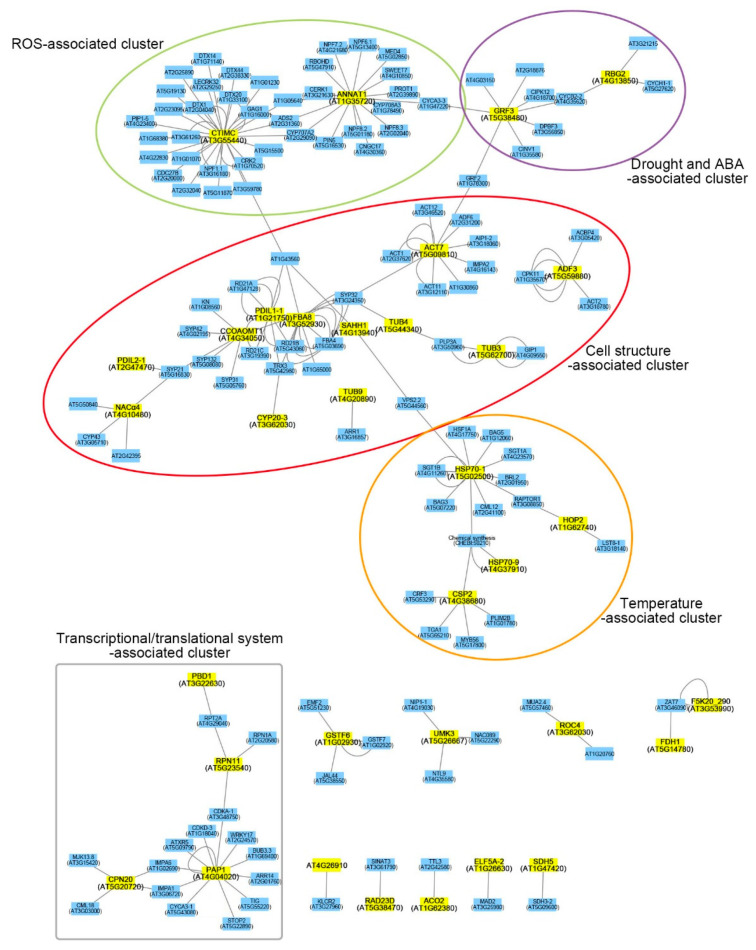
Interaction network analysis with DEPs by the Cytoscape software platform using IntAct molecular interaction database. Yellow squares indicate proteins of the input lists, whereas blue squares indicate proteins of physical interaction.

**Figure 5 ijms-22-05957-f005:**
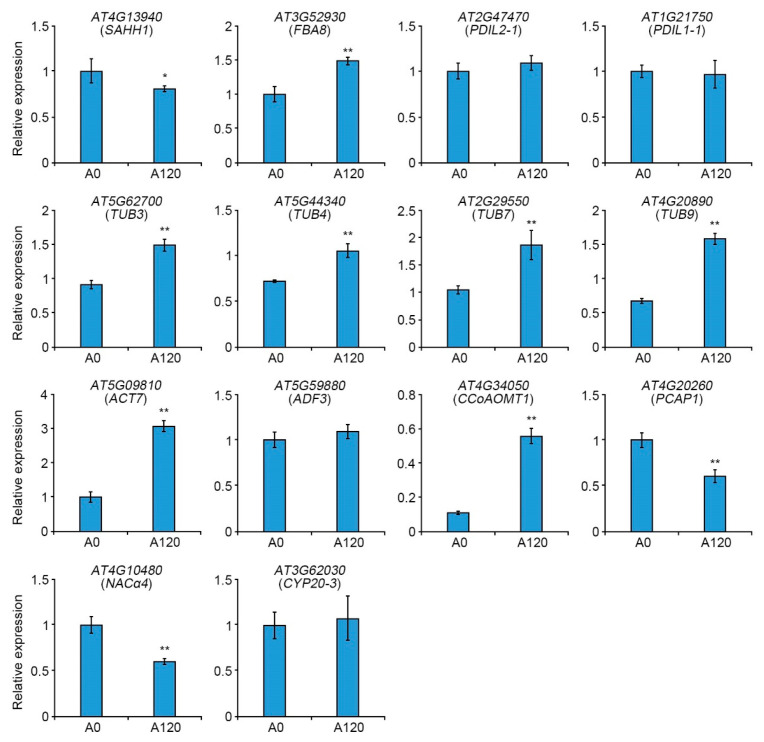
Transcript levels of cell structure-related genes in control (A0) and salt-adapted (A120) cells. Total RNAs were extracted from the A0 and A120 cells. Transcript levels were determined by quantitative real-time PCR (qRT-PCR). UBQ10 was used as a quantitative control for qRT-PCR. Error bars represent the standard deviation (SD) of three independent replicates. Asterisks indicate significant differences in the A0 cells (* *p*-value < 0.5; **, *p*-value ≤ 0.01, Student’s t-test).

**Figure 6 ijms-22-05957-f006:**
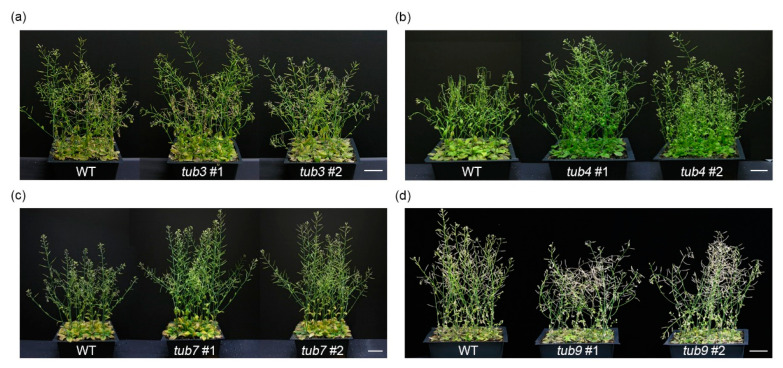
Characterization of loss-of-function mutants of *Arabidopsis* β-tubulin genes in salt stress. The 2-week-old wild-type (WT), tub3 (**a**), tub4 (**b**), tub7 (**c**), and tub9 (**d**) mutants grown on soil were treated with 130 mM NaCl for four weeks. Photos show a representative of 12 to 16 individual test plants. Scale bars indicate 5 cm.

**Figure 7 ijms-22-05957-f007:**
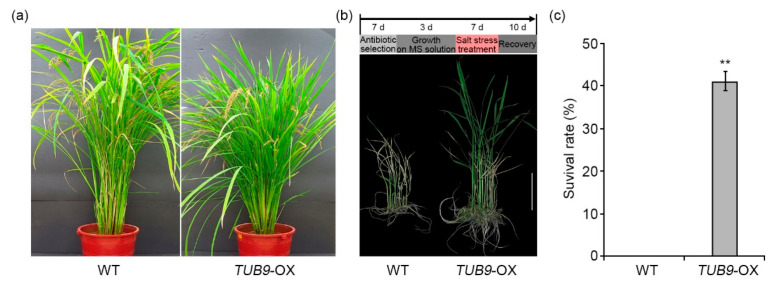
Characterization of *TUB9*-overexpressing transgenic rice plants. (**a**) Comparison of plant growth in wild-type (WT, “Ilmi” cultivar) and *TUB9*-overexpressing transgenic rice plants (*TUB9*-OX). Photographs were taken at maturity (milky ripening) stages. (**b**) Comparison of plant tolerance in WT and *TUB9*-OX plants to salt stress. Ten-day-old seedlings were treated with 120 mM NaCl for 7 days. After NaCl treatment, plants were recovered in MS medium without NaCl. Scale bars indicate 5 cm. (**c**) Quantitative analysis of survival rate of WT and *TUB9*-OX plants after salt stress treatment. Error bars represent the standard deviation (SD) of three independent replicates of the same experiment. Asterisks represent significant differences in the WT (** *p* ≤ 0.01, Student’s *t*-test).

**Table 1 ijms-22-05957-t001:** Identification of DEPs between control A0 and salt-adapted A120 *Arabidopsis* callus suspension-cultured cells.

Spot No. ^a^	Locus No.	Protein Name		Theo. Mr/pI ^b^	Queries Matched ^c^	Scores ^d^	Expect	Fold (A0 vs. A120)
1, 28	AT1G56070	LOS1	Low expression of osmotically responsive genes 1	95.10/ 5.89	31	566	3.70 × 10^−50^	1.554
2	AT1G62740	HOP2	Stress-inducible protein, putative	67.63/ 6.24	22	349	1.90 × 10^−28^	2.538
3	AT4G13940	SAHH1	S-adenosyl-L-homocysteine hydrolase 1	53.97/ 5.66	27	522	9.40 × 10^−46^	−1.133
4	AT1G51710	UBP6	Ubiquitin carboxyl-terminal hydrolase 6	54.00/ 5.82	27	467	3.00 × 10^−40^	2.473
5	AT4G01850	SAM-2	S-adenosylmethionine synthase 2	43.63/ 5.67	28	616	3.70 × 10^−55^	1.932
6	AT2G36880	MAT3	Methionine adenosyltransferase 3	42.93/ 5.76	33	699	1.90 × 10^−63^	1.249
7	AT1G77120	ADH1	Alcohol dehydrogenase class-P	41.84/ 5.83	27	798	2.40 × 10^−73^	−1.614
8	AT4G02930		GTP binding Elongation factor Tu family protein	49.61/ 6.25	33	1100	1.50 × 10^−103^	1.129
9	AT3G51800	CPR	Metallopeptidase M24 family protein	43.28/ 6.36	17	280	1.50 × 10^−21^	4.172
10	AT5G14780	FDH1	Formate dehydrogenase, chloroplastic/mitochondrial	42.67/ 7.12	25	560	1.50 × 10^−49^	2.048
11	AT4G26910		Dihydrolipoamide succinyltransferase	50.03/ 9.21	13	185	4.70 × 10^−12^	2.026
12, 50	AT3G04120	GAPC1	Glyceraldehyde-3-phosphate dehydrogenase 1, cytosolic	37.01/ 6.62	30	1040	1.50 × 10^−97^	4.510
13	AT5G43330	MDH2	Lactate/malate dehydrogenase family protein	35.98/ 7.00	24	675	4.70 * 10^−61^	1.407
14	AT5G23540	RPN11	26S proteasome non-ATPase regulatory subunit 14 homolog	34.39/ 6.31	18	241	1.20 × 10^−17^	5.121
15	AT3G52930	FBA8	Fructose-bisphosphate aldolase 8, cytosolic	38.86/ 6.05	31	1090	1.50 × 10^−102^	1.031
16	AT5G65020	ANNAT2	Annexin D2, calcium binding proteins	36.36/ 5.76	27	339	1.9 × 10^−27^	1.426
17	AT2G47470	PDIL2-1	Disulfide isomerase-like (PDIL) protein	39.81/ 5.80	20	576	3.70 × 10^−51^	1.308
18	AT5G02500	HSP70-1	*Arabidopsis thaliana* heat shock cognate protein 70-1	57.54/ 5.01	34	844	5.90 × 10^−78^	−4.195
19	AT1G21750	PDIL1-1	Disulfide isomerase-like (PDIL) protein	55.85/ 4.81	23	457	3.00 × 10^−39^	1.466
20, 21	AT5G62700	TUB3	Tubulin beta chain 3	51.27/ 4.73	38	778	2.40 × 10^−71^	3.032
22	AT2G29550	TUB7	Tubulin beta-7 chain	51.34/ 4.74	36	674	5.90 × 10^−61^	3.857
23	AT5G38470	RAD23D	Rad23 UV excision repair protein family	40.10/ 4.58	17	382	9.40 * 10^−32^	2.453
24	AT4G20890	TUB9	Tubulin beta-9 chain	50.31/ 4.69	38	715	4.70 × 10^−65^	1.649
25	AT5G44340	TUB4	Tubulin beta chain 4	50.36/ 4.76	31	490	1.50 ×× 10^−42^	2.158
26	AT4G37910	HSP70-9	Heat shock 70 kDa protein 9, mitochondrial	73.32/ 5.51	29	625	4.70 × 10^−56^	2.615
27	AT5G09810	ACT7	Actin 7	41.94/ 5.31	32	1100	1.50 × 10^−103^	1.110
29	AT1G35720	ANNAT1	Annexin D1, calcium binding proteins	36.30/ 5.21	27	754	5.90 × 10^−69^	2.469
30	AT1G79230	STR1	Thiosulfate/3-mercaptopyruvate sulfurtransferase 1, mitochondrial	42.15/ 5.95	21	529	1.90 × 10^−46^	1.487
31	AT3G53970		Probable proteasome inhibitor	32.15/ 4.94	15	329	1.90 × 10^−26^	3.353
32	AT1G62380	ACO2	1-aminocyclopropane-1-carboxylate oxidase 2	36.39/ 4.98	21	722	9.40 × 10^−66^	−2.900
33, 34	AT4G20260	PCAP1	Plasma membrane associated cation-binding protein 1	18.98/ 9.88	9	168	2.40 × 10^−10^	−1.309
37	AT5G38480	GRF3	14-3-3-like protein GF14 psi	32.00/ 4.91	8	149	1.90 × 10^−08^	1.052
38	AT4G04020	PAP1	Probable plastid-lipid-associated protein 1, chloroplastic	34.99/ 5.45	23	786	3.70 × 10^−72^	5.711
40	AT4G10480	NACα4	Nascent polypeptide-associated complex (NAC), alpha subunit family protein	23.10/ 4.25	8	202	9.40 × 10^−14^	2.031
41, 42	AT4G02450	P23-1	HSP20-like chaperones superfamily protein	25.38/ 4.46	10	248	2.40 × 10^−18^	3.315
43	AT4G34050	CCoAOMT1	S-adenosyl-L-methionine-dependent methyltransferases superfamily protein	29.25/ 5.13	20	659	1.90 × 10^−59^	7.222
45	AT1G47420	SDH5	Succinate dehydrogenase subunit 5, mitochondrial	28.15/ 6.19	15	410	1.50 × 10^−34^	5.433
46, 48	AT3G55440	CTIMC	Triosephosphate isomerase, cytosolic	27.38/ 5.39	26	904	5.90 × 10^−84^	1.155
47	AT5G20720	CPN20	20 kDa chaperonin, chloroplastic	26.79/ 8.86	8	87	0.033	1.638
49	AT5G26667	UMK3	P-loop containing nucleoside triphosphate hydrolases superfamily protein	22.58/ 5.79	13	298	2.40 × 10^−23^	2.088
51	AT1G02930	GSTF6	Glutathione S-transferase F6	23.47/ 5.80	18	467	3.00 * 10^−40^	1.795
52	AT3G22630	PBD1	Proteasome subunit beta type-2-A	22.64/ 5.95	15	138	2.40 * 10^−07^	1.952
53	AT4G38680	CSP2	Glycine-rich protein 2*, Arabidopsis thaliana* cold shock protein 2	19.49/ 5.62	5	97	0.0031	4.117
54	AT3G62030	CYP20-3 /ROC4	Peptidyl-prolyl cis-trans isomerase CYP20-3, chloroplastic	26.73/ 8.63	16	418	2.40 × 10^−35^	1.402
55	AT1G26630	ELF5A-2	Eukaryotic translation initiation factor 5A-1 (eIF-5A 1) protein	17.36/ 5.55	7	232	9.40 × 10^−17^	2.141
56	AT5G59880	ADF3	Actin-depolymerizing factor 3	16.03/ 5.93	10	545	4.70 × 10^−48^	1.290
57	AT3G53990	F5K20_290	Adenine nucleotide alpha hydrolases-like superfamily protein	17.90/ 5.66	21	575	4.70 × 10^−51^	1.502
58	AT3G23490	CYN	Cyanate hydratase	18.64/ 5.49	15	574	5.90 × 10^−51^	1.316
59	AT4G13850	RBG2	Glycine-rich RNA-binding protein 2, mitochondrial	14.74/ 6.73	7	268	2.40 × 10^−20^	13.161
60	AT5G18060	SAUR23	SAUR-like auxin-responsive protein family	72.78/ 5.87	21	69	1.9	2.814

^a^ The number of identification spots. ^b^ Theoretical mass (Mr, kDa) and pI of identified proteins. Theoretical values were retrieved from the protein database. ^c^ Number of matched peptides. ^d^ The mascot scores.

## Data Availability

The data presented in this study are available on request from the corresponding author.

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
