# Peer review of "Microtubule Dynamics Plays a Vital Role in Plant Adaptation and Tolerance to Salt Stress"

_ijms, 2021, doi:10.3390/ijms22115957_

Round 1
Reviewer 1 Report
Chun et al described the role of some tubulin genes under salt stress conditions in Arabidopsis mutant plants and in transgenic rice plants.
I have some major concerns for accepting the manuscript in IJMS journal.
- If Authors main aim to show the role of tubulin genes in slat stress conditions, they need to overexpress at least two tubulin genes in heterologous host. Even Tub4 also unregulated in qpcr experiment why they did not perform the overexpression of Tub4 in rice. Authors need to overexpress it.
- Authors need to show subcellular localization of wild and mutated Tubulin proteins (Tub4 & 9), which is important for microtubule dynamics.
- Authors did not show the microscopic root cellular structure of either Arabidopsis (mutant) or rice plants (OE) under normal and salt stress conditions. Scientific readers will see the comparison and role of tubulin genes for stabilizing the cellular structure under normal and slat stress conditions.(Not enough to show at suspension cell stage)
- Authors did not generate the complementary lines for Arabidopsis mutants (tub3,4,7,9). It’s important to generate the complementary lines for claiming the evidence.
- Authors should also check the phenotypic performance of Arabidopsis plants via overexpressing the important lignin biosynthesis gene CCoAOMT1
- Is there any reason to treat Arabidopsis and rice plants with different concentrations of NaCl. Its 120mM for rice and 130mM for Arabidopsis.
Author Response
For the response from the reviewer’s question and comments, we used blue letter for answers. And please, see the updated manuscript based on the reviewer’ comments. The changed sentences were marked with red in the new version of manuscript.
[Reviewer 1]
Chun et al described the role of some tubulin genes under salt stress conditions in Arabidopsis mutant plants and in transgenic rice plants.
I have some major concerns for accepting the manuscript in IJMS journal.
Overall Response to Reviewer 1 comments:
First of all, we appreciate the reviewer’s valuable comments on our manuscript. We absolutely agree with reviewer’s opinion and believe that these comments will improve the scientific quality of our manuscript.
As described in the manuscript, in this study, we mainly focused on understanding the molecular mechanisms underlying the adaptation of Arabidopsis cells to prolonged salt stress. For this purpose, we analyzed the changes of protein profiles between salt-adapted A120 cells and control A0 cells by using proteomics approach and suggested that the functions of TUB proteins are important in plant adaptation to long-term salt stress.
To confirm our proteomics results, we analyzed the phenotypes to salt stress of Arabidopsis mutants of corresponding genes to identified proteins from proteomics approach. From this molecular genetic analysis, we found that the tub9 mutant plants were more sensitive to salt stress than wild-type Col-0 plants, which was well correlated the results of its upregulation in mRNA and protein levels in response to salt stress, suggesting that TUB9 plays positive role in plant adaptation and tolerance to salt stress. Then, finally, we tested the possibility of its application in the development of salt tolerant crop plants by overexpressing Arabidopsis TUB9 gene in rice transgenic plants.
However, but interestingly, the tub4 mutant plants were more tolerant to salt stress than wild-type Col-0 plants, which was opposite to tub9 mutant phenotype, suggesting that TUB4 and TUB9 might play different roles in plant adaptation and tolerance to salt stress. Even though, in this study, we do not clarify their biological functions in plant response to salt stress, it is surely the next way to go for the further study. We are currently focusing on this research and hope to publish another paper.
Reviewer’s comments:
- If Authors main aim to show the role of tubulin genes in slat stress conditions, they need to overexpress at least two tubulin genes in heterologous host. Even Tub4 also unregulated in qpcr experiment why they did not perform the overexpression of Tub4 in rice. Authors need to overexpress it.
Response #1:
Thank you for your comment. As mentioned above, we generated rice transgenic plants overexpressing Arabidopsis TUB9 gene, because only TUB9 showed a nice correlation between mutant phenotype and mRNA and protein expression patterns under saline conditions. But, since we did not observe this correlation in other TUB genes, including TUB4 as well as TUB3 and TUB7, we thought that Arabidopsis TUB9 gene is the best candidate gene to use for generating salt tolerant crops. Thus we overexpressed TUB9 gene only in rice to enhance rice tolerance.
- Authors need to show subcellular localization of wild and mutated Tubulin proteins (Tub4 & 9), which is important for microtubule dynamics.
- Authors did not show the microscopic root cellular structure of either Arabidopsis (mutant) or rice plants (OE) under normal and salt stress conditions. Scientific readers will see the comparison and role of tubulin genes for stabilizing the cellular structure under normal and slat stress conditions.(Not enough to show at suspension cell stage)
Response #2 and #3:
Thank you for your comment. As suggested by reviewer, these results would be surely helpful for readers to understand the precise role of each TUB genes in changes of microtubule dynamics in plant response to salt stress, but these kinds of research are planning, partly performing currently, for the further study.
- Authors did not generate the complementary lines for Arabidopsis mutants (tub3,4,7,9). It’s important to generate the complementary lines for claiming the evidence.
Response #4:
Thank you for your comment. We agree that the confirmation of mutant lines by complementation assay is important to confirm the mutant phenotype. But, we tested many mutant lines corresponding to a number of identified proteins from proteomics, we could not perform this complementation assay, but instead, we confirmed that all tested mutant plants were RNA-null loss-of mutants by RT-PCR. However, in further study for analyzing the mode of action of each specific TUB gene in salt stress response, we need to confirm the mutant phenotypes by complementation assay, as suggested by reviewer.
- Authors should also check the phenotypic performance of Arabidopsis plants via overexpressing the important lignin biosynthesis gene CCoAOMT1.
Response #5:
Thank you for your comment. As mentioned above, we confirmed our proteomics results through the molecular genetic analysis using the Arabidopsis T-DNA insertion mutants. And, since we focused on TUB genes in this manuscript, we did not include the results of CCoAOMT1 gene. And, in our previous papers (Chun et al., 2019; Chun et al., 2021), we showed that the ccoaomt1 mutants are hypersensitive to not only salt stress (Chun et al., 2019), but also dehydration stress (Chun et al., 2021) compared to wild-type Col-0 plants. For CCoAOMT1 gene, we are planning to test whether overexpression of Arabidopsis CCoAOMT1 gene can enhance crop tolerance against multiple abiotic stresses in the further study.
Chun, H.J.; Baek, D.; Cho, H.M.; Lee, S.H.; Jin, B.J.; Yun, D.J.; Hong, Y.S.; Kim, M.C. Lignin biosynthesis genes play critical roles in the adaptation of Arabidopsis plants to high-salt stress. Plant Signal Behav. 2019, 14, e1625697, doi: 10.1080/15592324.2019.1625697.
Chun, H.J.; Lim, L.H.; Cheong, M.S.; Baek, D.; Park, M.S.; Cho, H.M.; Lee, S.H.; Jin, B.J.; No, D.H.; Cha, Y.J.; Lee, Y.B.; Hong, J.C.; Yun, D.J.; Kim, M.C. Arabidopsis CCoAOMT1 Plays a Role in Drought Stress Response via ROS- and ABA-Dependent Manners. Plants. 2021, 10, 831, doi: 10.3390/plants10050831.
- Is there any reason to treat Arabidopsis and rice plants with different concentrations of NaCl. Its 120mM for rice and 130mM for Arabidopsis.
Response #6:
Thank you for your comment. Before we conducted salt tolerance test with Arabidopsis and rice plants, we tested the various salt treatment conditions and identified the optimized conditions for salt tolerant test of Arabidopsis plants grown in soil and rice plants grown in hydroponic conditions. Thus, we conducted salt tolerance test with different assay methods for Arabidopsis and rice plants.

Reviewer 2 Report
The article of Hyun Jin Chun and coauthors is devoted to the study of the role of microtubules in plant adaptation to salt stress. Importance of β-tubulin proteins in plant adaptation to salt stress was shown by using 2-DE, RT-PCR, loss-of-function mutant plants and plants with gene overexpression. The data obtained in this paper are well presented and fully correspond to the topic of the special issue of the journal.
But I have some notes and questions:
1) Why were suspension-cultured cells from root callus used?
2) 42 line “to reestablish cellular homeostasis and protect proteins and cell membranes”
What is meant?
3) Salt concentration 120 mM NaCl was used in this paper for creating salt-adapted cells? What about effects of another salt concentrations?
4) Table 1. The height of the top row is less than the height of the text.
5) Figure 6. What is the difference between WT plants from the variants a,b,c,d?
6) 251-252 lines “…which indicated that Arabidopsis loss-of-function tub9 mutants were hypersensitive to salt stress than were WT plants..”
Needs to be formulated differently.
7) Why were transgenic rice plants overexpressing Arabidopsis TUB9 gene used? Why not Arabidopsis?
8) What the physiological role of TUB4 could be? Why is its expression increased in plants under stress conditions?
9) 286-289 lines “Results of GO and network analysis using proteomics data showed that most DEPs identified from salt-adapted cells were associated with regulating cell structures, including cytoskeleton and cell wall biogenesis (Figures 3 and 4).”
“most” needs to be replaced with “many”
10) 288-290 lines “Moreover, our gene expression and molecular genetic analyses revealed that β-tubulin family proteins function as positive and negative regulators in plant adaptation and tolerance to salt stress”
Needs to be replaced with “…β-tubulin family proteins play positive and negative role in….” as variant. What do these proteins regulate?
11) 300-302 lines “Functional network analysis revealed that the identified DEPs were included in various functional clusters, but mainly in cell structure-associated clusters, including cytoskeleton and cell wall biogenesis functions (Figure 4).”
“mainly” needs to be replaced with “many of them”
12) The “Discussion” section (3.1.1-3.1.5) contains a lot of literature data (total number of references is 89). In my opinion, these literary data and data of interaction network analysis are not of particular value without additional experiments.
13) The “Materials and methods” section does not contain RT-PCR method. How was it performed?
14) 480-481 lines “After 7 days, plants were recovered in MS solution without NaCl for 10 days.” Why is it done for WT seedlings and TUB9-OX rice plants?

Author Response
For the response from the reviewer’s question and comments, we used blue letter for answers. And please, see the updated manuscript based on the reviewer’ comments. The changed sentences were marked with red in the new version of manuscript.
[Reviewer 2]
The article of Hyun Jin Chun and coauthors is devoted to the study of the role of microtubules in plant adaptation to salt stress. Importance of β-tubulin proteins in plant adaptation to salt stress was shown by using 2-DE, RT-PCR, loss-of-function mutant plants and plants with gene overexpression. The data obtained in this paper are well presented and fully correspond to the topic of the special issue of the journal.
But I have some notes and questions:
Reviewer’s comments:
1) Why were suspension-cultured cells from root callus used?
Response #1:
Thank you for your comment. Plant roots play an important role in plant adaptation to salt stress, as the primary sensor. Thus, we generated Arabidopsis root callus suspension cultured cells adapted to prolonged salt stress as mentioned in our previous paper (Chun et al., 2019) to investigate the molecular mechanisms underlying plant adaptation to long-term salt stress using metabolomics approach. In this manuscript, we tried to analyze the changes of protein profiles in this salt-adapted cells using proteomics approach.
Chun, H.J. et al., Metabolic Adjustment of Arabidopsis Root Suspension Cells during Adaptation to Salt Stress and Mitotic Stress Memory. Plant Cell Physiol. 2019, 60, 612-625, doi: 10.1093/pcp/pcy231.
2) 42 line “to reestablish cellular homeostasis and protect proteins and cell membranes”
What is meant?
Response #2:
Thank you for your comment. We modified this part in the revised manuscript as following (lines 42-45);
“to reestablish cellular homeostasis, such as osmotic and ionic homeostasis, and protect proteins and cell membranes by using heat shock proteins (Hsps), chaperones, late embryogenesis abundant (LEA) proteins, osmoprotectants, and free-radical scavengers”
3) Salt concentration 120 mM NaCl was used in this paper for creating salt-adapted cells? What about effects of another salt concentrations?
Response #3:
Thank you for your comment. In our previous report, we generated the adapted cell lines to 120 mM NaCl by increasing NaCl concentration gradually (Chun et al., 2019). This salt-adapted cell line, named A120 (Adapted to 120 mM NaCl) cells, were able to grow and divide successfully in high salt growth conditions, thus we used this A120 cell line for further omics analyses, such as metabolomics and proteomics. In our previous paper (Chun et al., 2019), we showed that this A120 cells are more tolerant up to 200 mM NaCl concentrations compared to control cell line, A0.
Chun, H.J. et al., Metabolic Adjustment of Arabidopsis Root Suspension Cells during Adaptation to Salt Stress and Mitotic Stress Memory. Plant Cell Physiol. 2019, 60, 612-625, doi: 10.1093/pcp/pcy231.
4) Table 1. The height of the top row is less than the height of the text.
Response #4:
Thank you for your comment. We modified Table 1 accordingly in the revised manuscript.
5) Figure 6. What is the difference between WT plants from the variants a,b,c,d?
Response #5:
Thank you for your comment. We described the difference of phenotypes between WT plants and tub mutants in more detail in the revised manuscript as following (lines 251-256);
“The tub4 mutants displayed strongly tolerant phenotypes, such as enhanced plant height and late wilting of leaves, to salt stress compared to WT plants (Figure 6b). In contrast, tub9 mutants were hypersensitive to salt stress with a quickly wilting phenotype compared with WT plants (Figure 6d). Furthermore, tub3 and tub7 mutants showed similar phenotypes, including plant height and wilting of leaves, to WT plants under salt stress conditions (Figures 6a and 6c).”
6) 251-252 lines “…which indicated that Arabidopsis loss-of-function tub9 mutants were hypersensitive to salt stress than were WT plants..”
Needs to be formulated differently.
Response #6:
Thank you for your comment. We modified this sentence in the revised manuscript as following (lines 266-267);
“The hypersensitive phenotype of Arabidopsis tub9 mutants to salt stress suggests that the overexpression of Arabidopsis TUB9 gene can enhance crop tolerance to salt stress.”
7) Why were transgenic rice plants overexpressing Arabidopsis TUB9 gene used? Why not Arabidopsis?
Response #7:
Thank you for your comment. To confirm our proteomics results, we analyzed the phenotypes to salt stress of Arabidopsis mutants of corresponding genes to identified proteins from proteomics approach. From this molecular genetic analysis, we found that the tub9 mutant plants were more sensitive to salt stress than wild-type Col-0 plants, which was well correlated the results of its upregulation in mRNA and protein levels in response to salt stress, suggesting that TUB9 plays positive role in plant adaptation and tolerance to salt stress. Then, finally, we tested the possibility of its application in the development of salt tolerant crop plants by overexpressing Arabidopsis TUB9 gene in rice transgenic plants.
8) What the physiological role of TUB4 could be? Why is its expression increased in plants under stress conditions?
Response #8:
Thank you for your comment. As commented by reviewer, it would be interest to verified role of TUB4 in plant response to salt stress. The tub4 mutant plants were more tolerant to salt stress than wild-type Col-0 plants, which was opposite to tub9 mutant phenotype, suggesting that TUB4 and TUB9 might play different roles in plant adaptation and tolerance to salt stress and moreover, TUB4 might act as a negative regulator in plant tolerance to salt stress. Even though, in this study, we do not clarify their biological functions in plant response to salt stress, it is surely the next way to go for the further study.
Interestingly, it has been reported that gene expressions of many negative regulators in plant salt stress signaling pathway increase in response to salt stress as well as various stresses (Seok et al., 2017; Xu et al., 2019).
Seok, H.Y.; Woo, D.H.; Nguyen, L.V.; Tran, H.T.; Tarte, V.N.; Mehdi, S.M.; Lee, S.Y.; Moon, Y.H. Arabidopsis AtNAP functions as a negative regulator via repression of AREB1 in salt stress response. Planta. 2017, 245, 329-341. doi: 10.1007/s00425-016-2609-0.
Xu, Y.; Yu, Z.; Zhang, S.; Wu, C.; Yang, G.; Yan, K.; Zheng, C.; Huang, J. CYSTM3 negatively regulates salt stress tolerance in Arabidopsis. Plant Mol Biol. 2019, 99, 395-406. doi: 10.1007/s11103-019-00825-x.
9) 286-289 lines “Results of GO and network analysis using proteomics data showed that most DEPs identified from salt-adapted cells were associated with regulating cell structures, including cytoskeleton and cell wall biogenesis (Figures 3 and 4).”
“most” needs to be replaced with “many”
Response #9:
Thank you for your comment. We changed “most” to “many” in the revised manuscript (line 303).
10) 288-290 lines “Moreover, our gene expression and molecular genetic analyses revealed that β-tubulin family proteins function as positive and negative regulators in plant adaptation and tolerance to salt stress”
Needs to be replaced with “…β-tubulin family proteins play positive and negative role in….” as variant. What do these proteins regulate?
Response #10:
Thank you for your comment. We changed “that β-tubulin family proteins function as positive and negative regulators in plant adaptation” to “that β-tubulin family proteins play positive and negative role in plant adaptation” in the revised manuscript (lines 305-307).
Concerning the biological functions of each Arabidopsis TUB protein in salt stress response; As commented by reviewer, it would be very interesting and important to verify the roles of β-tubulin family proteins in plant adaptation and tolerance to salt stress, but unfortunately, we did not clarify this in this manuscript. However, it is surely the next way to go for the further study.
11) 300-302 lines “Functional network analysis revealed that the identified DEPs were included in various functional clusters, but mainly in cell structure-associated clusters, including cytoskeleton and cell wall biogenesis functions (Figure 4).”
“mainly” needs to be replaced with “many of them”
Response #11:
Thank you for your comment. We changed “mainly” to “many of them” in the revised manuscript (line 319).
12) The “Discussion” section (3.1.1-3.1.5) contains a lot of literature data (total number of references is 89). In my opinion, these literary data and data of interaction network analysis are not of particular value without additional experiments.
Response #12:
Thank you for your comment. We reorganized and removed many of references from interaction network analysis parts of Discussion in the revised manuscript (lines 322-417).
13) The “Materials and methods” section does not contain RT-PCR method. How was it performed?
Response #13:
Thank you for your comment. We included the methods of qRT-PCR analysis in the Materials and Methods part of the revised manuscript (lines 485-497).
14) 480-481 lines “After 7 days, plants were recovered in MS solution without NaCl for 10 days.” Why is it done for WT seedlings and TUB9-OX rice plants?
Response #14:
Thank you for your comment. To investigate the recuperative ability of TUB9 overexpression rice plants from salt stress, salt-treated WT and TUB9-OX plants were recovered in normal MS medium without additional NaCl. And this salt tolerance assay method has been used in many previous papers testing the salt tolerance of rice plants (Xiang et al., 2013; Wang et al., 2019).
Xiang. J.; Ran. J.; Zou. J.; Zhou. X.; Liu. A.; Zhang. X.; Peng. Y.; Tang. N.; Luo. G.; Chen. X. Heat shock factor OsHsfB2b negatively regulates drought and salt tolerance in rice. Plant Cell Rep. 2013, 32, 1795-1806. doi: 10.1007/s00299-013-1492-4.
Wang. X.; Zhang. H.; Shao. L.-Y.; Yan. X.; Peng. H.; Ouyang. J.-X.; Li. S.-B. Expression and function analysis of a rice OsHSP40 gene under salt stress. Genes Genomics. 2019, 41, 175-182. doi: 10.1007/s13258-018-0749-2.

Round 2
Reviewer 1 Report
Dear Authors,
I am not satisfied with your responses. The MS needs subcellular localaization experiments. Without some experimental proofs, i cant accept this MS to publish.
Author Response
I agree with that this subcellular localization test will provide a valuable information for the verification of the functions of tubulin protein family in salt adaptation process. In fact, we are planning to test the changes of microtubule dynamics in response to salt stress and the roles of tubulin family members, such as TUB4 and TUB9, in this plant response by using cell biological approach.
For this experiment, we need to generate transgenic plant expressing TUA6::GFP fusion construct, which can show the overall subcellular microtubule dynamics (Celler et al., Methods Mol Biol. 2016 ; 1365: 155–184. doi:10.1007/978-1-4939-3124-8_8.), and then test the changes of overall cellular microtubule dynamics in TUA6::GFP transgenic plants under with and without salt stress conditions. Moreover, to test the specific role of each tubulin family protein in this process, we need to transform TUA6::GFP construct into each tub gene mutant plants, such as tub4 and tub9 mutants, then repeat the experiment as describe above. But, these experiments require much time, at least more than a year, so that, we cannot provide these data in a short time. Thus, we want to publish these data in our next paper. I hope you can understand this situation.